# Management of Traumatic Brain Injury in Patients with DOAC Therapy–Are the “New” Oral Anticoagulants Really Safer?

**DOI:** 10.3390/jcm11216268

**Published:** 2022-10-25

**Authors:** Anna Antoni, Lukas Wedrich, Martin Schauperl, Leonard Höchtl-Lee, Irene K. Sigmund, Markus Gregori, Johannes Leitgeb, Elisabeth Schwendenwein, Stefan Hajdu

**Affiliations:** 1Department of Orthopedics and Trauma-Surgery, Medical University of Vienna, 1090 Vienna, Austria; 2Department of Orthopaedics and Traumatology, Klinik Floridsdorf, Vienna Healthcare Group, 1210 Vienna, Austria; 3Department of Emergency Medicine, Klinik Ottakring, Vienna Healthcare Group, 1160 Vienna, Austria; 4Department of Orthopaedics and Traumatology, University Hospital Tulln, 3430 Tulln an der Donau, Austria

**Keywords:** traumatic brain injury, antithrombotic therapy, DOAC, NOAC, direct oral anticoagulants

## Abstract

(1) Background: In recent years, “new” direct oral anticoagulants (DOAC) have gradually replaced other antithrombotic therapies. The international literature agrees on the increased mortality for traumatic brain injury (TBI) patients using vitamin K antagonists (VKA), but thus far, there are insufficient data on the influence of DOAC on the outcome of TBI. (2) Methods: We retrospectively analyzed data from all patients who presented with head trauma using antithrombotic therapy. Outcome parameters were the presence of pathologies on the initial CT, occurrence of delayed intracranial hemorrhage, surgical intervention, and death. (3) Results: In total, data of 1169 patients were reviewed. Of those, 1084 (92.7%) had a mild TBI, 67 (5.7%) moderate TBI, and 17 (1.5%) severe TBI. In total, 456 patients (39%) used DOAC and 713 patients (61%) used VKA, antiplatelet therapy, or prophylactic doses of low molecular weight heparin at the time of trauma. The groups showed no significant differences in age, injury mechanisms, or GCS at presentation. Overall, the initial cranial CT showed pathologies in 85 patients (7.3%). Twenty-five patients with head trauma and DOAC therapy had pathological findings on CT (5.5%), 11 patients with VKA (4.8%), and 48 patients with antiplatelet therapy (10.6%). There was a statistically significant difference in occurrence of CT pathologies between DOAC alone compared to acetylsalicylic acid (4.9 vs. 10.5%, *p* = 0.04). Delayed intracranial hemorrhage after an initially negative CT during in-hospital observation occurred in one patient (0.2%) in the DOAC group, two patients (0.9%) in the VKA group, and four patients (0.9%) in the antiplatelet group without statistical significance. Head trauma related surgery was performed in three patients (0.7%) in the DOAC group, two patients (0.9%) in the VKA group, and six patients (1.3%) in the antiplatelet group without statistical significance. Death due to head trauma occurred in four patients (0.9%) of the DOAC group compared to one patient (0.4%) of the VKA group and five patients (1.1%) of the antiplatelet group without statistical significance. (4) Conclusions: Our data suggest a comparable risk of pathological CT findings, delayed intracranial hemorrhage, surgical interventions, and death after blunt head trauma for patients with DOAC compared to VKA, but a lower risk for pathological CT findings compared to platelet inhibitors. As VKA are known to increase mortality, our data suggest that similar caution should be used when treating patients with head trauma and DOAC, but the overall numbers of serious or severe courses after simple falls remain low. We recommend routine CT for all head trauma patients with antithrombotic therapy but the role of in-hospital observation for patients with mild TBI remains a matter of debate.

## 1. Introduction

In recent years, “new” oral anticoagulants (NOAC), or later referred to as direct oral anticoagulants (DOAC), have gradually replaced other antithrombotic therapies (ATT) for the prevention and treatment of venous thromboembolism. The benefits of DOAC are a higher convenience for patients and lower risk of life-threatening bleeding compared to vitamin K antagonists (VKA) [1,2,3]. However, even after more than ten years of widespread DOAC use, there is still controversy about the influence of DOAC on the outcomes of traumatic brain injury (TBI).

The increased mortality of TBI for patients using VKA is well described in the literature. Large studies showed an increased mortality for patients with VKA after head trauma [4,5,6,7] but there is conflicting evidence for all other types of ATT [8,9,10,11].

While the convenience for patients and reduced risk of life-threatening bleeding led to an increase in DOAC prescriptions, the trauma community repeatedly expressed concerns due to the limited availability of laboratory tests, reversal agents, and lack of outcome data for trauma patients [12,13]. Studies showed a higher risk for early and delayed intracranial hemorrhage and increased mortality after head trauma for DOAC compared to VKA patients [14,15], while on the other hand, other studies showed a lower risk and lower mortality [16,17,18,19,20,21]. Over the years, more and more studies described a comparable risk of intracranial hemorrhage and mortality for patients with DOAC when compared to VKA [22,23,24].

### Aim of the Study

The aim of the study was to compare the incidence of pathologies on the initial computer tomography, occurrence of delayed intracranial hemorrhage, surgical interventions, intensive care admissions, and death in patients with head trauma using either DOAC, Vitamin K antagonists, platelet inhibitors, or prophylactic doses of low molecular weight heparin.

## 2. Materials and Methods

We retrospectively analyzed data from patients using antithrombotic therapy who presented at our level I trauma department with blunt head trauma, either with neurological symptoms (mild TBI, moderate TBI, severe TBI) or without neurological symptoms (head trauma/minimal TBI) caused by blunt forces from a fall, traffic accident, or direct blow to the head.

Inclusion criteria were all degrees of blunt head trauma with or without neurological symptoms. Patients had to be at least 18 years old and received an initial computer tomography (CT) and ensuing in-hospital observation for at least 24 h.

Alcohol intoxication at presentation and an age below 18 years constituted exclusion criteria.

Patients using DOAC at time of injury, between January and December 2017, were compared to our traumatic brain injury database, which included blunt head trauma patients using either Vitamin K antagonists (VKA), platelet inhibitors, or prophylactic doses of low molecular weight heparin (LMWH) between January and December 2013.

The DOAC investigated were Dabigatran (Pradaxa^®^ Boehringer Ingelheim International GmbH, Ingelheim am Rhein, Germany), Apixaban (Eliquis^®^ Pfizer Inc., New York, NY, USA), Edoxaban (Lixiana^®^ Daiichi Sankyo Co., Tokyo, Japan), and Rivaroxaban (Xarelto^®^ Bayer Vital GmbH, Leverkusen, Germany). The VKA used was Marcumar^®^ (Viatris Inc., Canonsburg, PA, USA), the antiplatelet medications, acetylsalicylic acid (ASA, Aspirin^®^ Bayer Vital GmbH, Leverkusen, Germany) and Clopidogrel (Plavix^®^ Sanofi, Bridgewater, MA, USA). Additional antithrombotic therapies included LMWH in prophylactic doses as well as combination therapies of the above medications.

The largest subgroups were patients with DOAC only (without combination therapies), VKA only, antiplatelet therapy (ASA, Clopidogrel, and the combination therapy of ASA and Clopidogrel), and ASA only. These groups were used for statistical analysis, while the other subgroups were only mentioned descriptively due to their small size. As the study period predates routine implementation of andexanet alfa in our department, we did not include reversal agents in our data analysis.

Outcome parameters were presence of pathologies on the initial CT, occurrence of delayed intracranial hemorrhage (DIH), surgical interventions, intensive care admission, duration of hospital stay, and death.

Clinical data were retrieved from the local hospital information system (patient data, trauma mechanism, neurological status, radiologic findings, laboratory tests, outcome parameters) and anonymously recorded in our database.

The study complies with authorization by the local Ethics Committee (1745/2018).

At the time, clinical protocol required an initial CT and neurologic observation for a minimum of 24 h for all patients with head trauma and any antithrombotic therapy. In the control group, an additional follow-up CT was performed after 24 to 48 h of in-hospital observation for patients with VKA or Clopidogrel, despite a negative initial CT. This protocol was no longer used during data acquisition for the DOAC group, and the indication for follow-up CTs was based on individual clinical decisions in case of neurologic symptoms’ onset during the observation period. We amended the policy to reflect the results of our previously published head trauma database analysis [25].

Our department routinely withdraws the ATT during in-hospital observation, with the exception of ASA, or in case of a high risk for thromboembolic events. Reversal agents are administered only in selected cases of moderate or severe TBI and were not recorded for this study due to the low number and retrospective design of the study.

In our level I trauma department, trauma surgeons routinely treat and operate the majority of acute TBI and request assistance from neurosurgeons as required. Until 2015 trauma surgery was a specialization in Austria that included the surgical training for neurotrauma. Based on this qualification, TBI is still treated by trauma surgeons who completed the “historical” trauma surgery training in Austria. In recent years there has been a constant shift towards operative treatment of TBI by multi-disciplinary teams or by neurosurgeons.

Statistical analysis of the retrospective data was performed descriptively. Metric variables were compared between the DOAC and other antithrombotic therapy groups by calculating the mean, the range, and standard deviation. Other variables were compared by frequency. The Fisher’s exact test was used to compare the risk of outcome parameters for the different antithrombotic therapy groups with Bonferroni-adjusted alpha for multiple testing. The *p*-value was set at α < 0.05.

## 3. Results

Overall, 1169 patients with head trauma consistent with the aforementioned criteria were included. Of these, 456 patients (39%) used DOAC (referred to as DOAC group) and 713 patients (61%) in the control group used vitamin K antagonists (VKA), antiplatelet therapy, or prophylactic doses of low molecular weight heparin (LMWH) at the time of trauma (Table 1).

The groups showed no significant differences in age, sex, injury mechanisms, or Glasgow Coma Score (GCS) at presentation. The mean age of the whole study cohort was 82 years. Of these, 647 patients (55.3%) were female. The mean GCS was 15. Classified by the Abbreviated Injury Scale (AIS) for head trauma, 1084 patients (92.7%) had a mild traumatic brain injury (AIS 1), 67 (5.7%) a moderate TBI (AIS 2-3), and 17 (1.5%) a severe TBI (AIS ≥ 4). Mechanisms of trauma were a simple fall (*n* = 1104; 94.4%), traffic accidents (*n* = 18; 1.5%), a fall from great height (*n* = 12; 1.0%), and sport injuries (*n* = 11; 0.9%). In 24 patients, the mechanism of trauma was unknown. Patients with moderate and severe TBI had a simple fall as injury mechanism in 63 cases (75%). In the moderate and severe TBI group, only two patients were classified as polytrauma with an ISS greater than 15.

In total, 736 cases (63.0%) had visible signs of trauma to the head (such as hematoma, abrasions, or lacerations).

The initial cranial CT showed pathologies in 85 patients (*n* = 85/1169; 7.3%): 25 in the DOAC group (*n* = 25/456; 5.5%) and 60 in the control group (*n* = 60/713; 8.4%). There was a statistically significant difference in occurrence of CT pathologies between DOAC alone and antiplatelet therapy (4.9 vs. 10.6%, *p* = 0.02) as well as DOAC alone compared to ASA (4.9 vs. 10.5%, *p* = 0.04) (Table 2).

Delayed intracranial hemorrhage (DIH) after an initially negative CT during in-hospital observation occurred in seven patients (*n* = 7/1169; 0.6%). The detailed results of the subgroups are shown in Table 2. There were no statistically significant differences between the subgroups (*p* > 0.05). It must be emphasized that patients with VKA and Clopidogrel (part of the antiplatelet therapy group) had a routine control CT after 24 to 48 h, while all other patients had a follow up CT only in cases of clinical deterioration during the in-hospital observation. There were no DIH related deaths in our study population. Details of the patients with DIH are shown in Table 3. The patient with DIH and Rivaroxaban presented with an occipital fracture on his initial CT. The patient with decompressive craniectomy after DIH had a second fall during the in-hospital observation period with a consecutive hemorrhage.

There were no statistically significant differences (*p* > 0.05) between all subgroups regarding head trauma related surgery (decompressive craniectomy, burr holes), which was performed in 12 patients overall (*n* = 12/1169; 1.0%). Three surgeries were needed in the DOAC group (0.7%) and nine surgeries in the control group (*n* = 9/1169; 1.3%) (Table 2). Details of patients who underwent surgery are shown in Table 4.

Intensive care unit admission was necessary in 16 patients in total (*n* = 16/1169; 1.4%). In the DOAC group, six patients required ICU treatment (*n* = 6/456; 1.3%) compared to ten in the control group (*n* = 10/713; 1.4%). The detailed results for the subgroups are illustrated in Table 2, but there were no statistically significant differences between them (*p* > 0.05).

The mean hospital stay was 3.7 days in the DOAC group and 4 days in the control group among surviving patients, without statistically significant difference (*p* > 0.05).

Overall mortality was 1.5% (18/1169) in all included patients admitted for head trauma using ATT. Death after moderate or severe TBI occurred in four patients (0.9%) in the DOAC group compared to six patients (0.8%) in the control group. The detailed numbers shown in Table 2 did not yield statistically significant differences in TBI mortality between the different antithrombotic therapies (*p* > 0.05).

## 4. Discussion

With our results, we are able to contribute to the ongoing debate about the management of patients sustaining head trauma during antithrombotic therapy.

In our study, there were no significant differences regarding incidence of intracranial hemorrhages or other parameters, such as surgical interventions, ICU admission, length of hospital stay, or TBI-related death between patients in the DOAC and in the VKA group.

Since the increased mortality for patients with head trauma and VKA therapy is well known, we conclude that DOAC use is not safer than VKA for patients with head trauma, but the overall numbers of serious or severe courses after simple falls remain low.

Non-trauma related complications of DOAC therapy are described as lower compared to VKA with comparable antithrombotic efficacy [1,2,3]. For DOAC use and trauma in general, Maung et al. showed that VKA had a higher overall mortality compared to DOAC with similar incidence of TBI in their subgroups [26].

Studies on TBI repeatedly showed either lower or higher mortality for DOAC patients compared to VKA [14,15,16,17,18,19,20,21], but in recent years several published studies support our conclusion of comparable risks for these patients [22,23,24].

Surprisingly, there is still controversy about the influence of antiplatelet therapy on mortality after head trauma. Our study from 2019 showed a low rate of delayed intracranial hemorrhage for patients using ASA with 0.5% of all admitted patients with blunt head trauma and an initially negative CT [25]. McMillian et al. and van den Brand et al. showed higher rates of intracranial hemorrhages after head trauma when patients used antiplatelet therapy similar to our results [8,11], but there is still a lack of other large-scale studies to show the influence of antiplatelets on the outcome of these patients [9,10]. The recent systematic review and meta-analysis of Mathieu et al. showed an increase in ICH progression and neurosurgical interventions for patients with dual antiplatelet therapy, but not for ASA alone [27].

Consequently, the other important finding of our study is the statistically significant difference of occurrence of CT pathologies between DOAC alone and antiplatelet therapy (4.9 vs. 10.6%, *p*= 0.02), as well as DOAC alone compared to ASA (4.9 vs. 10.5%, *p*= 0.04). This may suggest that DOAC use is safer than antiplatelet therapy in cases of head trauma.

Although there is a widespread consensus that head trauma patients with ATT should routinely undergo CT imaging [28,29,30,31,32] and some study groups recommend a short period of hospitalization for these patients [30,31,33], the need for in-hospital observation of neurologically intact patients with negative CT findings is still a matter of debate [33,34].

Cost effective strategies are en vogue, however, studies show that increasing CT scanning is associated with higher inpatient hospital admissions, especially in a group of patients older than 65 years [35].

Since physicians are more and more confronted with geriatric patients at emergency departments, minor head trauma has become a leading reason for hospital admissions in many countries. It must be recognized that in-hospital observations of patients suffering dementia can be associated with complications, such as adverse drug events [36], in-hospital falls, and delirium [37,38]. In a retrospective multi-center analysis, Verschoof et al. reported that none of 905 anticoagulated patients developed delayed intracranial hemorrhage within 24 h, therefore, the authors concluded that “routine hospitalization seems unwarranted” [32].

The benefit of in-hospital observation is lacking evidence and studies should be performed if adverse events overweigh the potential benefits, as this might influence current practices. The Austrian interdisciplinary consensus statement recommends discharging patients under ASA monotherapy with a negative CT scan and GCS 15 [30], but suggests in-hospital observation for all other ATT. Considering the results of our present study, it should be discussed if GCS 15 patients with a negative CT, independently of any ATT, may be discharged from hospital due to the low numbers of DIH.

The major strength of this study is the great number of patients with mild TBI. However, limitations are the retrospective design, as well as the differing clinical protocols during the collection of DOAC patients in 2017 and other ATT in 2013, when patients with VKA and Clopidogrel had a routine follow-up CT during in-hospital observation. Due to this fact, the low and not significantly different DIH numbers may be biased towards slightly lower rates in the DOAC group.

Due to the low number of severe TBI, we cannot make any valid conclusions about hemorrhage progression and neurological outcomes. The lack of clinical outcome parameters, such as the GOSE, is another disadvantage: Nevertheless, it would be of limited scientific value for the main group of minimal or mild traumatic brain injury in our study population.

A follow-up study with even larger case numbers to compare results of different DOAC types, especially the “xabans” (Apixaban, Edoxaban, Rivaroxaban) and Dabigatran, and outcome data for severe TBI after use of DOAC reversal agents which were not used in this study population at the time of investigation, is currently planned in our department.

## 5. Conclusions

Our data suggests a comparable risk of pathological CT findings, delayed intracranial hemorrhage, surgical interventions, and death after blunt head trauma for patients with DOAC compared to VKA, but a lower risk for pathological CT findings compared to platelet inhibitors. As large studies have published the VKA increase in mortality, our data suggests that patients with head trauma and DOAC warrant similar caution. Additionally, the relatively higher risk of intracranial hemorrhage for head trauma patients with antiplatelet therapy in our study mandates caution for patients with head trauma and antiplatelets.

We recommend routine CT for all head trauma patients with antithrombotic therapy but the role of in-hospital observation for patients with mild TBI remains a matter of debate.

## Figures and Tables

**Table 1 jcm-11-06268-t001:** Types of antithrombotic therapies in the DOAC group and in the control group.

Type of ATT	Number of Patients (% of Subgroup)
DOAC group:	456
Apixaban (Eliquis^®^)	152 (33.3%)
Edoxaban (Lixiana^®^)	21 (4.6%)
Dabigatran (Pradaxa^®^)	55 (12.1%)
Rivaroxaban (Xarelto^®^)	200 (43.9%)
DOAC and antiplatelet	28 (6.1%)
Control group:	713
VKA(Marcumar^®^)	229 (32.1%)
ASA (Aspirin ^®^)	351 (49.2%)
Clopidogrel (Plavix ^®^)	88 (12.3%)
ASA and Clopidogrel	13 (1.8%)
VKA and ASA	12 (1.7%)
LMWH	20 (2.8%)

Abbreviations: ATT: antithrombotic therapy, DOAC: direct oral anticoagulants, VKA: vitamin K antagonists, ASA: acetylsalicylic acid, LMWH: low molecular weight heparin.

**Table 2 jcm-11-06268-t002:** Results of outcome parameters in the subgroups. For statistical purposes, the groups consist of DOAC alone without other ATT combination therapies, VKA alone without combination therapies, all types of antiplatelet therapy including dual therapy and ASA alone.

	DOAC Alone	VKA Alone	Antiplatelet Therapy	ASA Alone
	*n* (% of Subgroup)	*n* (% of Subgroup)	*n* (% of Subgroup)	*n* (% of Subgroup)
Pathological first CT	21 (4.9%)	11 (4.8%)	48 (10.6%)	37 (10.5%)
DIH	1 (0.2%)	2 (0.9%)	4 (0.9%)	2 (0.6%)
Surgery	3 (0.7%)	2 (0.9%)	6 (1.3%)	6 (1.7%)
ICU	6 (1.3%)	0	9 (2.0%)	7 (2.0%)
TBI death	4 (0.9%)	1 (0.4%)	5 (1.1%)	3 (0.9%)

Abbreviations: DOAC: direct oral anticoagulants, ATT: antithrombotic therapy, VKA: vitamin K antagonists, ASA: acetylsalicylic acid, CT: computer tomography, DIH: delayed intracranial hemorrhage, ICU: intensive care unit, TBI: traumatic brain injury.

**Table 3 jcm-11-06268-t003:** Details of patients with delayed intracranial hemorrhage including history of unconsciousness or amnesia at presentation, GCS at presentation, visible head trauma, such as wounds or hematoma, neurological symptoms during the in-hospital observation, and surgery due to DIH.

Patient	Antithrombotic Therapy	Unconsciousness/Amnesia	GCS	Visible Head Trauma	Neurological Symptoms	DIH	Surgery
m, 88y	Rivaroxaban	yes	15	yes	no	SDH, ICH	no
m, 85y	ASA	yes	15	yes	yes	SDH, SAH	decompressive craniectomy
f, 82y	ASA	no	15	yes	no	SDH	no
f, 79y	Clopidogrel	yes	14	yes	no	SDH	no
f, 90y	ASA, Clopidogrel	no	15	yes	no	SDH	no
f, 82y	Vitamin K antagonist	no	15	yes	no	SAH	no
m, 93y	Vitamin K antagonist	no	15	yes	no	SAH	no

Abbreviations: GCS = Glasgow Coma Scale, DIH = delayed intracranial hemorrhage, ASA = acetylsalicylic acid, SDH = subdural hematoma, ICH = intracerebral hematoma, SAH = subarachnoid hematoma.

**Table 4 jcm-11-06268-t004:** Details of patients who underwent head trauma related surgery including history of unconsciousness or amnesia at presentation, GCS at presentation, visible head trauma such as wounds or hematoma, type of hemorrhage, and type of TBI surgery.

Patient	Antithrombotic Therapy	Unconsciousness/Amnesia	GCS	Visible Head Trauma	Hemorrhage	Surgery	TBI Related Death
f, 78y	Dabigatran	unknown	10	yes	SDH	decompressive craniectomy	yes
f, 78y	Apixaban	no	15	yes	SDH, SAH	decompressive craniectomy	no
m, 91y	Rivaroxaban	no	15	yes	SDH, SAH	burr holes	no
m, 44y	ASA	unknown	8	yes	ICH	decompressive craniectomy	no
f, 88y	ASA	no	15	yes	SDH	decompressive craniectomy	yes
m, 85y	ASA	yes	15	yes	SDH, SAH	decompressive craniectomy	no
f, 96y	ASA	no	15	yes	SDH, ICH	burr holes	no
f, 92y	ASA	yes	15	yes	SDH	burr holes	no
m, 88y	ASA	yes	15	yes	SDH, SAH	burr holes	no
m, 81y	Vitamin K Antagonist	no	15	yes	SDH, SAH	burr holes	no
f, 82y	Vitamin K Antagonist	no	15	yes	SDH	burr holes	no
m, 76y	Heparin	no	15	no	SDH	burr holes	no

Abbreviations: GCS = Glasgow Coma Scale, TBI = traumatic brain injury, ASA = acetylsalicylic acid, SDH = subdural hematoma, SAH = subarachnoid hematoma, ICH = intracerebral hematoma.

## Data Availability

The data presented in this study are available on request from the corresponding author. The data are not publicly available due to Ethics Committee regulations.

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
