# Peer review of "Management of Traumatic Brain Injury in Patients with DOAC Therapy–Are the “New” Oral Anticoagulants Really Safer?"

_jcm, 2022, doi:10.3390/jcm11216268_

Round 1

Reviewer 1 Report

Overall it is a well written study and helpful for us all who treat TBI routinely. My main are of criticism are the following:

1. I am not sure why a neurosurgeon did not participate in this study. There must be neurosurgeons in this trauma center working with the author groups. This conspicuous absence creates questions such as: did the neurosurgeons have any concerns about the methodology and decided not to participate?

I am a neurosurgeon and obviously biased in this review, but these type of studies are better done in a multi-disciplinary group, where each participant brings in her/his own expertise. 

2. Why not perform a logistic regression and control for all variables? The chosen methodology is not the most powerful to control for confounders.

3. A better description of the patients who had delayed intracranial bleeding and also of those who underwent surgery is needed. After all, this whole study is about these subjects who had the outcome of interest. What kind of intracranial bleeding they had? what kind of surgery? what was the EBL? 

Author Response

  1. “I am not sure why a neurosurgeon did not participate in this study. There must be neurosurgeons in this trauma center working with the author groups. This conspicuous absence creates questions such as: did the neurosurgeons have any concerns about the methodology and decided not to participate?

I am a neurosurgeon and obviously biased in this review, but these type of studies are better done in a multi-disciplinary group, where each participant brings in her/his own expertise.”

Until 2015 trauma surgery was a specialization in Austria that included the surgical training for neurotrauma. Based on this qualification, TBI is still treated by trauma surgeons who followed the “historical” trauma surgery specialization in Austria. Naturally, in recent years there has been a constant shift towards operative treatment of TBI by multi-disciplinary teams or by neurosurgeons.

Our department for trauma surgery still routinely treats and operates the majority of acute TBI and requests assistance from neurosurgeons as required. Based on these facts, we hope that you do understand the lack of neurosurgeons in our study team and added this information to the methods section. Be assured that based on the recent changes in the Austrian trauma surgery training, we strive to cooperate more closely with our neurosurgery department clinically as well as scientifically.

  1. “Why not perform a logistic regression and control for all variables? The chosen methodology is not the most powerful to control for confounders.”

The reply of our statistician was, that the number of events per variable were too low to calculate a logistic regression analysis.

  1. A better description of the patients who had delayed intracranial bleeding and also of those who underwent surgery is needed. After all, this whole study is about these subjects who had the outcome of interest. What kind of intracranial bleeding they had? what kind of surgery? what was the EBL? 

Thank you for pointing this out, we included more information on patients with delayed intracranial bleeding and surgery in the results.

Reviewer 2 Report

The authors present a relatively large study (patients > 1000) on antithrombotic agents and the risk to develop post-traumatic intracranial hematomas. The study is overall well-written and concerns a relevant clinical topic. Some comments below.

- I was quite surprised that ASA was associated with such a high rate of pathological CT. E.g. a previous systematic review showed that APs are at least not associated with hemorrhage progression (doi.org/10.1089/neur.2022.0042) Please discuss this a bit more.

- I still think it would be valuable if you provide data on how you dealt with the antithrombotic agents after head trauma. Withdrawal alone? Reversal - how and when? This is important in order to understand the effect on delayed hematomas, the need/risk for surgery and mortality.

- Please specify the setting (trauma center) in the Abstract/Methods.

Author Response

- “I was quite surprised that ASA was associated with such a high rate of pathological CT. E.g. a previous systematic review showed that APs are at least not associated with hemorrhage progression (doi.org/10.1089/neur.2022.0042) Please discuss this a bit more.”

Thank you for pointing that out, since we too were surprised about these results. We added the named study to the discussion.

- “I still think it would be valuable if you provide data on how you dealt with the antithrombotic agents after head trauma. Withdrawal alone? Reversal - how and when? This is important in order to understand the effect on delayed hematomas, the need/risk for surgery and mortality.”

We added this information to the methods section and to the discussion, since this is an important part of the clinical management of these patients, thank you for emphasizing this.

- “Please specify the setting (trauma center) in the Abstract/Methods”.

We added additional information in the methods about the trauma system in Austria and procedures in our department for better understanding of the setting.

Round 2

Reviewer 2 Report

The authors have addressed my concerns.